# Atherogenic Index of Plasma Predicts Obstructive Coronary Artery Disease in Patients with Stable Angina Pectoris

**DOI:** 10.3390/diagnostics13203249

**Published:** 2023-10-19

**Authors:** Haci Ali Kurklu, Turkan Seda Tan, Nil Ozyuncu, Emir Baskovski, Cagdas Ozdol

**Affiliations:** 1Department of Cardiovascular Medicine, Ankara Etlik Research Hospital, Ankara 06590, Turkey; hacialikurklu@gmail.com; 2Department of Cardiovascular Medicine, Ankara University School of Medicine, Ankara 06590, Turkey; nilozyuncu@yahoo.com (N.O.); emirbaskovski@yahoo.com (E.B.); cagdasozdol@gmail.com (C.O.)

**Keywords:** coronary artery disease, cardiovascular risk factor, atherogenic index of plasma, coronary artery stenosis

## Abstract

Aims: Chronic coronary syndrome is associated with several risk factors, such as dyslipidemia and hypertension. The atherogenic index of plasma (AIP) has been demonstrated to be a biochemical risk factor for coronary artery disease (CAD). This study aimed to determine whether the AIP is an effective parameter for estimating obstructive CAD. Methods and Results: A total of 345 patients (with a mean age of 62.2 ± 10.3; 63% male) who underwent coronary angiography were included in this study. Obstructive CAD is defined as having one or more vessels with a stenosis level of ≥50%. Depending on the presence of obstructive CAD, all patients were divided into two groups. The mean AIP value was found to be 0.538 ± 0.26 in the study group. The AIP values were significantly higher in the obstructive coronary artery group (AIP; 0.49 ± 0.26 vs. 0.58 ± 0.27, *p* = 0.002). According to a univariable analysis, AIP values were significantly associated with obstructive coronary artery disease [OR: 3.74 (CI 95% 1.62–8.64), *p* = 0.020]. The AIP was further adjusted for confounding risk factors in three multivariable analysis models and, all three models showed a significant association. According to an ROC analysis, 0.49 is the cut-off value for AIP, and a value above 0.49 indicates 50% coronary artpery stenosis. Conclusions: The AIP may be used in the assessment of cardiovascular risk for patients with stable angina pectoris, and it may also be used to estimate obstructive CAD.

## 1. Introduction

Coronary artery disease (CAD), also called ischemic heart disease, is one of the most common causes of mortality and morbidity worldwide [1]. Atherosclerotic plaques in the coronary arteries are the main feature of this disease. CAD can be acute or chronic according to its onset and duration and categorized as either an acute coronary syndrome (ACS) or a chronic coronary syndrome (CCS) [2,3]. The most common clinical forms of CCS are stable angina and/or dyspnea, newly onset heart failure, and/or left-ventricular dysfunction. Additionally, other forms of CSS occur in asymptomatic and symptomatic patients who had ACS within one year or patients with recent revascularization in one year and/or >1-year revascularization. Moreover, patients suspected of having vasospastic or microvascular disease as well as those with CAD who are detected at screening comprise the other forms of CCS [3].

CCS risk factors including a family history of CVD, dyslipidemia, metabolic syndrome, diabetes, hypertension, and smoking have been well established in recent guidelines, and risk stratifications for CAD have been created using these factors [4]. Particularly, dyslipidemia has been highly investigated, and it is thought that lipid-lowering treatment is the main risk modifier for CAD. Dyslipidemia is defined as an increase in low-density lipoprotein cholesterol (LDL-C), total cholesterol, and triglyceride (TG) and a reduction in high-density lipoprotein cholesterol (HDL-C). Recent guidelines focus on using LDL-lowering treatments to decrease CAD risk [4,5]. Nevertheless, about half of the residual cardiovascular risk remains even when LDL-C levels are lowered to the recommended levels. The atherogenic index of plasma (AIP) has been demonstrated as a new biochemical risk indicator for CAD development as a result of several laboratory findings [6]. Calculated using the formula log (TG/HDL-C), the AIP is a measure of plasma atherogenicity based on a positive correlation between cholesterol esterification rates, remnant lipoproteinemia, and lipoprotein particle size. The AIP is a powerful predictor of atherosclerosis and coronary heart disease, and it can accurately represent the link between protective and atherogenic lipoproteins [7].

In this study, we investigated whether the AIP could predict CAD among patients with stable angina pectoris. In addition, we sought to determine the AIP as a non-invasive parameter for estimating obstructive coronary artery disease.

## 2. Materials and Methods

Patient Data. A total of three hundred ninety-four patients with suspected CAD and who had undergone coronary angiography were retrospectively reviewed. Three hundred forty-five of those who had coronary angiography images and laboratory results were included in our study. The medical histories, including all clinical and demographic data including CAD risk factors such as hypertension (HT), diabetes mellitus (DM), dyslipidemia, smoking, family history, and body mass index, were obtained from electronic medical records. None of the patients had been previously diagnosed with CAD or ACS. In the study population, 247 patients had stable angina with typical symptoms that were refractory to medical treatment, and 98 patients had atypical symptoms. Laboratory results were received within 24 h before coronary angiography. The cholesterol values were measured using fasting blood samples obtained within 12 h of fasting. Patients with acute coronary syndrome, unstable coronary artery disease, previous coronary stent implantation and established coronary artery disease, severe valvular diseases, malignancies, cardiomyopathies, thyroid diseases, and familial hypercholesterolemia were excluded from our study. The study protocol was reviewed and approved by an ethical committee.

Coronary angiography: Coronary angiography was performed in accordance with the standard procedure by an interventional cardiologist. We used femoral artery or radial artery access and obtained standard left anterior oblique (LAO) and right anterior oblique (RAO) projections with cranial and caudal angulations for the assessment of the left coronary artery and LAO and RAO projections for the assessment of the right coronary artery [8,9].

Quantitative coronary angiography (QCA) calculation: QCA was performed according to a standard protocol, in which the maximum stenosis was determined from two orthogonal views. The angiograms were read by two blinded expert interventional cardiologists. The angiograms were carefully evaluated, and QCAs were conducted on any possible obstructions by the primary reader. A secondary reader ensured the quality and accuracy of the QCA. A patient with obstructive CAD is defined as having stenosis of 50% or greater in one or more vessels [10,11]. 

AIP calculation: Triglyceride (TG) and high-density lipoprotein levels were obtained from the laboratory results, which were obtained within 24 h before a coronary angiogram. The AIP is a logarithmically converted ratio of TG to HDL-C in a molar concentration (millimoles per liter). AIP was calculated as follows: log^10^ (TG/HDLC) [12].

### Statistical Analysis

Baseline characteristics are presented as the mean ± SD for continuous variables, were compared using Student’s *t*-test, or as percentages for categorical variable differences, were compared using the chi-square test. A *p*-value < 0.05 was defined as statistically significant Univariate and multivariate analyses based on the logistic regression model were performed to identify the relationship between cardiovascular risk factors and coronary artery stenosis. In order to avoid multicollinearity, parameters with a strong correlation with AIP (TG, HDL-C r > 0.7) were not entered into the multivariable analysis. Other variables in the univariable analysis were entered into the multivariable analysis. Parameters with *p*-value < 0.05 in multivariable analysis were defined as predictors of coronary artery stenosis. Using the receiver operating characteristic (ROC) curves based on the logistic regression method, a cutoff value of AIP was also found. All data were analyzed using IBM SPSS Statistics version 26 (SPSS Inc., Chicago, IL, USA).

## 3. Results

Baseline characteristics. Three hundred forty-five patients who underwent coronary angiography with a suspicion of CCS (mean age of 62.2 ± 10.3; 63% male) were included in this study. A diagnosis of obstructive coronary artery disease was defined as 50% or greater stenosis in at least one coronary artery. Having one or more vessels with a stenosis value of 50% or greater was defined as obstructive coronary artery disease, and all patients were divided into two groups based on whether or not they had obstructive coronary artery disease. The ≥50% group (190 patients; 55%) was defined as the obstructive coronary artery group, and the <50% group (155 patients, 45%) was defined as the non-obstructive coronary artery group. Among those with obstructive coronary arteries, 32 patients had left main coronary artery (LMCA) stenosis, 154 patients had left anterior descending artery [13] stenosis, 112 patients had circumflex artery (CX) stenosis, and 126 patients had right coronary artery stenosis [8]. Additionally, 36% (*n* = 124) of the patients in the obstructive coronary artery group had multi-vessel coronary artery stenosis. Among the groups, we compared demographic and clinical characteristics, laboratory results, and medication use (Table 1). The obstructive coronary artery group was older (age; 60.6 ± 10.5 vs. 63.2 ± 10.1, *p* = 0.002) and contained a higher percentage of males [male 85 (55.5%) vs. 152 (80%), *p* < 0.0001]. There were no differences in medication use. Similarly, the laboratory results (obtained within 24 hr. prior to coronary angiography), including complete blood counts, creatine levels, glomerular filtration ratios (GFR), and fasting plasma glucose (FPG) levels, were also similar between the groups. Further, the LDL-C levels (LDL-C = 114.1 ± 36 vs. 108.3 ± 36.1, *p* = 0.2) and TC levels (TC = 180.8 ± 41.3 vs. 186.8 ± 48.9, *p* = 0.23) were not different between the two groups. In contrast, the high-density cholesterol level (HDL-C; 43.4 ± 11.3 vs. 40.4 ± 11.4, *p* = 0.01) was significantly lower, and TG (TG; 147.0 ± 78.5 vs. 168.7 ± 100.9, *p* = 0.03) was significantly higher in the group with obstructive coronary arteries. The risk factors for coronary artery disease, such as diabetes mellitus and family history, were similar between the two groups, with the exception of HT, which was significantly more prevalent in the group with obstructive coronary artery disease [HT; 78 (50.3%) vs. 120 (63.2%), *p* = 0.02].

Atherogenic Index of Plasma: The AIP calculated via log (TG/HDL-C) was significantly higher in the obstructive coronary artery group (AIP; 0.49 ± 0.26 vs. 0.58 ± 0.27, *p* = 0.002) (Table 1).

Univariate and multivariate analyses of parameters associated with ≥50% coronary artery stenosis. We entered the parameters in Table 1 with a *p*-value of <0.05 into the univariable and multivariable logistic regression models to identify how these variables are associated with obstructive coronary artery disease. Due to the strong correlation (r > 0.7) between the TG, HDL-C, and AIP values (Appendix A), we did not include them in multivariable analysis.

In order to evaluate their relationship with coronary artery occlusion, variables with *p* < 0.05 in Table 1 (age, gender, BMI, HT, TG, HDL-C, LDL-C, and AIP) were included in the univariable analysis. Univariable modeling (Table 2) revealed that there were significant relationships between AIP values and obstructive coronary artery disease [OR: 3.74 (CI 95% 1.62–8.64), *p* = 0.002]. In addition, age [OR: 1.02 (CI 95% 1–1.04), *p* = 0.04] and gender [OR: 3.2 (CI 95% 1.99–5.16), *p* < 0.000] were also predictors of obstructive coronary artery disease. Having hypertension [OR: 1.69 (CI 95% 1.09–2.6), *p* = 0.02] and an increased body mass index value (BMI kg/cm^2^) [OR: 1.15 (CI 95% 1.0–1.33), *p* = 0.04] were highly associated with coronary artery stenosis. Furthermore, higher TG levels [OR: 1.31 (CI 95% 1.02–1.7), *p* = 0.03] and lower HDL-C levels [OR: 0.98 (CI 95% 0.96–1.0), *p* = 0.02], which are the parameters used to calculate the AIP, were predictors of obstructive coronary artery disease. However, the LDL-C levels were similar between the two groups as a result of statin use [OR: 0.99 (CI 95% 0.99–1.0), *p* = 0.2]. Consequently, no associations were found between LDL-C levels and obstructive coronary artery disease (Table 2).

The univariate analysis revealed that all the parameters were individually associated with coronary artery occlusion. Therefore, each parameter was entered individually into the multivariate analysis in order to determine the most accurate predictor of coronary artery occlusion. There was a significant correlation between AIP values and coronary artery occlusion,. Consequently, the AIP was adjusted for confounding risk factors in a multivariate analysis in order to evaluate its impact on coronary artery occlusions. Each classical risk factor was added to the multivariable analysis individually. Following this, we developed three separate models to analyze the relationship between the AIP and coronary artery occlusion (Table 3). Model 1 was adjusted for age and gender; it revealed a significant association between AIP values and obstructive coronary artery disease [OR: 3.39 (CI 95% 1.41–8.13), *p* = 0.003]. Based on the results of model 1, age [OR: 1.0 (CI 95% 1.01–1.06), *p* = 0.0001] and gender [ODR3.53 (CI 95% 2.1–5.9), *p* = 0.0000] were also independent predictors of coronary artery occlusion. Model 2 was adjusted for BMI and LDL-C in addition to Model 1, and Model 3 was adjusted for HT in addition to the confounders of Model 2. The association between AIP values and coronary artery occlusion remained significant under Model 2 [OR: 2.97 (CI 95% 1.2–7.39) *p* = 0.02]. Furthermore, although age [OR: 1.04 (CI 95% 1.02–1.06), *p* < 0.001] and gender [OR: 3.56 (CI 95% 2.12–5.98), *p* < 0.0000] were found to be independent predictors of coronary artery occlusion However, BMI [OR: 1.16 (CI 95% 0.99–1.35), *p* = 0.06] and LDL-C [OR: 0.99 (CI 95% 0.99–1.0, *p* = 0.4)] were not associated with coronary artery occlusion in the multivariable analysis of Model 2.

Additionally, in Model 3, the AIP was highly correlated with coronary artery occlusion [OR: 2.76 (CI 95% 1.1–6.91) *p* = 0.03]. Moreover, age [OR: 1.04 (CI 95% 1.01–1.06), *p* = 0.006] and gender [OR: 3.6 (CI 95% 2.17–6.18), *p* < 0.000] were strongly correlated with coronary artery occlusion, whereas HT [OR: 0.68 (CI 95% 0.42–1.13), *p* = 0.1], LDL-C [OR: 0.99 (CI 95% 0.99–1.0, *p* = 0.5)], and BMI [OR: 1.15 (CI 95% 0.99–1.35), *p* = 0.06] did not correlate with coronary artery occlusion in Model 3. As a result of adjusting for classical risk factors separately in all three models, AIP values have been identified as a significant predictor of coronary artery occlusion.

### ROC Analysis

In the present study, we investigated the individual effects of the parameters used to calculate AIP and found that HDL-C had lower accuracy [AUC: 0.41 (CI 95%: 0.35–0.47; *p* = 0.06), specificity 52%, and sensitivity 37%] than TG [AUC: 0.56 (CI 95%: 0.50–0.62; *p* = 0.04), specificity 55%, and sensitivity 54%]. Although TG and HDL-C were used to calculate the AIP, it was found that the AIP had a higher accuracy for estimating obstructive coronary artery disease [AUC: 0.60; CI 95%: 0.53–0.65; *p* = 0.002), specificity 51% and sensitivity 62%]. The cut-off value for the AIP was 0.49, and an AIP value above 0.49 was estimated to correspond to 50% coronary artery stenosis (Figure 1).

## 4. Discussion

The results of this study confirm that higher TG levels and lower HDL-C levels are significantly associated with coronary artery disease. The major finding of this study was that AIP values (the log of TG/HDL-C) were higher in patients with CAD. Further, the AIP was found to be a significant predictor of CAD even after adjusting for confounding risk factors. In addition, AIP values showed a better correlation with CAD than the parameters that were used in its corresponding formula (HLD-C and TC).

Cardiovascular disease still represents the greatest burden of disease, with high mortality and morbidity rates [1,2]. In this regard, cardiac risk stratification is essential for improving preventive and therapeutic measures [4]. The main causes and modifiable risk factors of atherosclerosis are blood apolipoprotein-B-containing lipoproteins, high blood pressure, smoking, adiposity, and diabetes mellitus. Additionally, there are a number of other relevant risk factors and clinical conditions, such as gender, age, and ethnicity. These factors are taken into account when estimating an individual’s cardiovascular risk [14]. In accordance with previous European Society of Cardiology (ESC) guidelines, the Systematic Coronary Risk Estimation (SCORE) method was developed and updated in the 2021 guidelines to evaluate 10 years of CVD risk estimation based on several of the risk factors mentioned above [14,15]. In estimating CVD risk, LDL-C is one of the most commonly used and emphasized variables, so LDL-C-lowering treatments are typically designed primarily to prevent CVD [5].

In the presence of endothelial dysfunction, small TG-rich lipoproteins and their remnant particles, known as ApoB-containing lipoproteins, have a tendency to cross the endothelial barrier, resulting in lipid deposition and atherosclerosis. The SCORE risk algorithm utilizes non-high-density lipoprotein cholesterol (non-HDL-C) values to estimate coronary heart disease risk, which includes all atherogenic lipoproteins (apo-B contained), such as LDL-C and TG, and is calculated as follows: Non-HDL-C = TC − HDL-C. Non-HDL-C can be used to identify the apo-B-containing proteins that are highly associated with atheroma formation. Therefore, elevated plasma TG levels indicate an increase in ApoB-containing proteins and, consequently, an increase in the risk of atherosclerotic cardiovascular disease (ASCVD) [5,14]. As a result of this information, TG levels and plasma LDL levels can be evaluated in order to determine whether a lipid-lowering treatment reduces the risk of atherogenicity. Therefore, a treatment can be adjusted by taking into account the TG value in addition to the LDL-C value.

Several biomarkers associated with CAD have been identified over the past few years, including inflammatory biomarkers such as C-reactive protein [16] and fibrinogen [17] and lipid-related biomarkers such as lipoprotein-associated phospholipase A2 [18] and lipoprotein A [19]. Nevertheless, recent guidelines do not recommend the use of biomarkers in risk stratification since they could result in confusion in risk assessment. In the recent guidelines, ApoB analysis is recommended as a tool for risk assessment, particularly for patients with high TG levels, diabetes, obesity, metabolic syndrome, or very low LDL-C levels. According to the 2019 ESC/EAS Guidelines for the management of dyslipidemia regarding lipid modification to reduce cardiovascular risk, ApoB measurements are recommended as a class I indication instead of LDL-C for primary screening, diagnosis, and management. Furthermore, ApoB may be used instead of non-HDL-C for individuals with high TG, diabetes, obesity, or very low LDL-C levels.

The atherogenic index of plasma (AIP) is a novel biomarker that includes a logarithmically transformed ratio of triglycerides to high-density lipoprotein (HDL) cholesterol in molar concentrations [20]. Additionally, the AIP can be calculated using only a standard lipid profile, making it an easily accessible biomarker [21]. Despite the lack of widespread availability and the limited cost-effectiveness of ApoB measurement, AIP measurement is an easy and inexpensive method, and it is not accompanied by additional costs beyond cholesterol measurement. Due to the fact that the AIP is calculated using TG, which is an ApoB-containing lipoprotein, it may indirectly provide information regarding the amount of ApoB. We believe that using the AIP along with LDL-C values may be an effective method for estimating ApoB.

It is well known that a low HDL level, a component of the AIP, is an independent risk factor for coronary artery disease. Moreover, as a biomarker, HDL-C is an effective tool for refining SCORE2 risk estimation. Additionally, TG, an ApoB-containing lipoprotein that is highly associated with ASCVD (mentioned above), is another component of the AIP. Moreover, a major benefit of the AIP is its consideration of hypertriglyceridemia and low HDL levels as independent markers of coronary artery disease in order to enhance their predictive value [4,6,21,22]. As a matter of fact, the AIP’s correlation with lipoprotein particle size most likely explains the nature of the relationship between AIP and CVD incidence [21]. There is an inverse relationship between the diameter of LDL-C and the AIP, with the latter being a substitute for minute, dense LDL particles. Thus, an increase in AIP values indicates that oxidized particles are more likely to produce foamy cells, resulting in an increase in LDL-C and oxidized apoprotein B combinations, which have been shown to be highly atherogenic. The overexpression of adhesion molecules and the activation of oxygen radicals have been directly linked to endothelial dysfunction due to the promotion of lipid peroxidation via high AIP values [22]. Furthermore, HDL-C is a component of the AIP. It transports cholesterol from peripheral tissues to the liver and contains antioxidant enzymes [23]. Clinical studies have confirmed these theoretical findings by demonstrating a strong association between AIP and carotid artery intima–media thickness [23], arterial stiffness [24], and coronary artery calcification [25,26].

Metabolic syndrome (MetS) is a group of cardiometabolic risk factors (CMR). There is no doubt that MetS is a leading cause of type 2 diabetes mellitus (T2DM) and cardiovascular disease (CVD), which are still among the leading causes of morbidity and mortality, as well as some of the most prevalent healthcare concerns. Recently, numerous studies have concluded that certain CMR factors have a positive relationship with the atherogenic index of plasma (AIP) [27,28,29]. Moreover, studies have shown that AIP values below 0.11 are associated with low CVD risk, whereas values between 0.11 and 0.21 as well as those greater than 0.21 are associated with intermediate and greater CVD risk, respectively [28,29]. An analysis of 32 articles revealed that a large waist circumference (WC), high triglyceride (TG) levels, high levels of insulin resistance (IR), and low high-density lipoprotein cholesterol (HDLC) concentrations were strongly correlated with increased AIP values. There were a few studies that examined blood pressure (BP), and the results were inconsistent [30]. Based on these studies, it can be concluded that the AIP is associated with WC, TG, IR, and HDL-C. There is no clear evidence linking AIP to blood pressure. Due to the retrospective nature of our study, we were not able to obtain WC and IR values from the patients; however, the BMI values of the patients with obstructive coronary artery disease were higher. In spite of the fact that the corresponding BMI values do not satisfy the definition of obesity, the increased AIP values and BMI values in this group support the findings of this study. Considering all of the findings presented above, it is reasonable to conclude that an increased AIP value indicates an increased risk for metabolic syndrome among patients, which eventually leads to an increase in the risk of cardiovascular disease (CVD).

A number of studies have found a link between AIP values and the progression of patients with acute coronary syndromes and myocardial infarctions. A Turkish study has demonstrated that AIP values are independently associated with a complete lack of reflow following primary percutaneous coronary intervention among patients with ST-elevated myocardial infarctions [31]. According to another study, post-myocardial-infarction patients with lower AIP values (0.24%) exhibited almost four-times-higher hospital mortality than those with higher AIP values [32]. Another study examining the relationship between acute coronary syndrome and AIP values among patients under 35 years of age found that AIP was independently associated with the presence and severity of coronary artery disease among young patients [33]. According to our study, there was a correlation between AIP values and obstructive coronary artery disease. There were no ST-T changes detected in the electrocardiogram (ECG), and no troponin elevations were observed in the patients. Therefore, acute coronary syndrome was excluded. However, due to the lack of an opportunity to analyze the patients’ plaque structures, plaque vulnerability could not be determined. The relationship between AIP values and vulnerable plaque needs to be evaluated in further research.

In many studies, the relationship between AIP, major cardiovascular events and prognosis has been investigated. For instance, the ACCORD study, using a large-scale analysis, demonstrated that higher AIP values were an independent predictor of survival among patients with Type 2 DM [34]. In addition, some studies have shown that higher AIP values are highly associated with major cardiovascular events (MACEs) among patients with and without diabetes [11,16,21,23]. Furthermore, Khosravi et al. found that AIP values were an independent biomarker that can differentiate unstable from stable plaques with 89.70% sensitivity and 34% specificity [35]. The AIP has also been found to be associated with CAD severity in some studies. For instance, Mangalesh et.al. found that AIP values were highly associated with both MACE within 3 years and the severity of CAD among patients with established coronary artery disease detected via coronary computed tomography angiography [36]. Moreover, Balci et. al. demonstrated that increased plasma AIP values were strongly associated with decreased FFR values in chronic coronary syndrome patients with intermediate coronary artery stenosis [37]. Another study analyzing a large population noted a strong correlation between increased AIP values and in-stent restenosis [38]. According to these studies, coronary artery disease severity is generally assessed among patients who already have coronary artery disease. However, in our study, we sought to determine whether or not AIP could be used to detect obstructive coronary artery disease in patients with suspected chronic coronary syndromes. As a consequence, we were able to demonstrate a strong association between AIP and obstructive coronary artery disease. Furthermore, risk was further increased by higher levels of AIP. According to these findings, the AIP may be an important marker for the early detection of this disease. Therefore, the AIP may be used as a biomarker to investigate coronary artery stenosis in order to improve the efficiency of non-invasive imaging techniques. Our study found that although the LDL-C levels were similar between the two groups of patients, the TG levels were significantly higher and the HDL-C levels were significantly lower in the patients with obstructive coronary artery disease. Furthermore, the level of AIP was significantly elevated in the group with obstructive coronary artery disease. Based on this information, it can be concluded that LDL-lowering treatments should be regulated in accordance with both TG and LDL levels in order to achieve optimum results. As a result of our findings, LDL-lowering treatments may be adjusted in accordance with the AIP cut-off value to minimize CVD risk by evaluating the decrease in AIP values, as LDL values, since AIP values were more closely correlated with the presence of obstructive coronary arteries than TG and HDL values in our study.

### Study Limitations

There are several limitations to our study, including the fact that it was a single-center study of a retrospective nature. The second limitation was that we were unable to determine how the other risk factors, such as diabetes mellitus, hypertension, and smoking, affected plaque burden because we did not know their onsets or durations of exposure. Furthermore, due to the retrospective nature of our study. Additionally, we were unable to determine the durations of the lipid lowering treatments for the study population. Also, the LDL-C levels indicated that appropriate dose titration was not performed. Despite presenting cardiovascular risk factors that warrant lipid-lowering treatment, some patients fail to receive it. A number of these factors, however, contribute to this study’s limitations. Furthermore, in our study, high TG and low HDL-C levels were the main contributors to coronary occlusion, while the LDL-C values were similar. In light of these findings, we concluded that lipid-lowering therapy might be titrated and dosed based on both LDL-C and AIP values. Prospective studies with a larger number of patients will be essential to demonstrate the efficacy of the AIP in the estimation of obstructive coronary artery disease.

## 5. Conclusions

The atherogenic index of plasma, which is calculated as log^10^ (TG-c/HDL-c), has been identified as a coronary artery risk factor in several studies. Moreover, a high AIP value is associated with a poor prognosis among patients with coronary artery disease. This study confirmed that the AIP is an independent predictor of CAD among patients suspected of having CCS. We also observed that increased AIP values are highly related to obstructive stenosis. Therefore, we believe that the AIP can provide an additional contribution to the CVD risk algorithm. In addition, a reduction in AIP values, including the lowering of LDL levels, may also be an important component of lipid-lowering therapies.

## Figures and Tables

**Figure 1 diagnostics-13-03249-f001:**
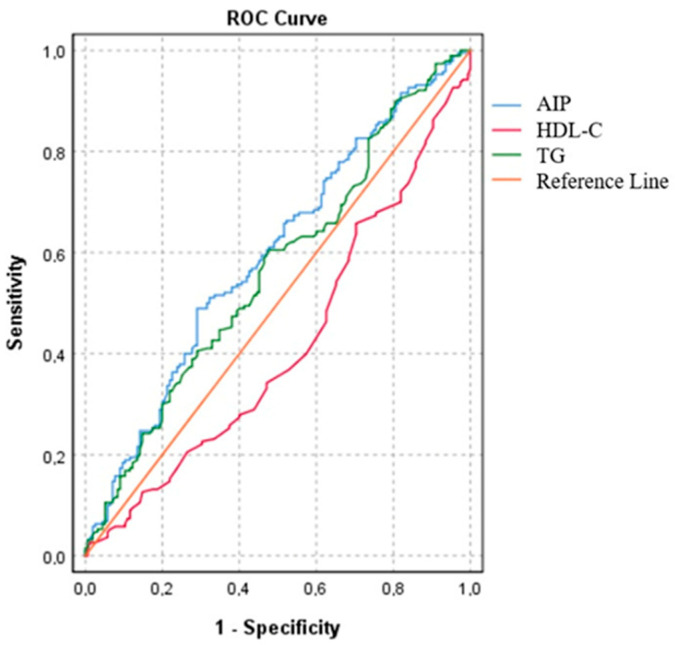
Receiver operating curves of estimated obstructive coronary artery disease. Cut-off point of 0.49% for AIP AUC = 0.60; (CI 95%: 0.53–0.65); *p* = 0.002.

**Table 1 diagnostics-13-03249-t001:** Baseline characteristics divided by ≥50% coronary artery stenosis.

Characteristics	<50% Coronary Stenosis Group (*n* = 155)	≥50% Coronary Stenosis Group (*n* = 190)	*p* Value
Age, year	60.6 ± 10.5	63.2 ± 10.1	0.02
Gender			
Male%	85 (55.5)	152 (80)	<0.0001
Female%	69 (44.5)	38 (20)	<0.0001
BMI kg/cm^2^	23.1 ± 1.5	23.5 ± 1.51	0.04
HT (%)	78 (50.3)	120 (63.2)	0.02
DM (%)	57 (36.8)	63 (33.2)	0.48
HL (%)	35 (22.6)	51 (26.8)	0.36
Smoking %	70 (45.2)	103 (54.2)	0.09
Family History %	49 (31.6)	65 (34.2)	0.61
Medication			
ACE inhibitors (%)	69 (44.5)	86 (45.3)	0.9
ARB (%)	17 (11)	21 (11)	1
Beta Blocker (%)	133 (70)	118 (76)	0.2
Any dihydropyridine	42 (21.1)	56 (29.5)	0.60
Statin (%)	68 (43.9)	71 (37.4)	0.20
Aldosterone inhibitors (%)	31 (20)	28 (14.7)	0.2
Diuretic (%) *	79 (51)	109 (57.4)	0.23
Metformin	63 (40.6)	61 (32.1)	0.1
Any SGLT2 inhibitor	61 (39.4)	69 (36.3)	0.6
Insulin	48 (31)	56 (29.5)	0.8
Laboratory Result			
FPG mg/dL	121.3 ± 52.9	112 ± 44.37	0.07
Hemoglobin g/dL	13.7 ± 2.1	14.1 ± 1.7	0.93
Leukocyte	8.5 ± 2.5	8.6 ± 2.4	0.65
Neutrophil	5.25 ± 2.1	7.5 ± 19.8	0.18
Lymphocyte	2.4 ± 2	2.4 ± 1.5	0.9
Platelet	256 ± 71.5	256 ± 66	0.9
Creatine (mg/dL)	0.86 ± 0.35	0.94 ± 0.38	0.06
GFR	86.3 ± 19.6	83.6 ± 18.9	0.2
TC, mg/dL	180.8 ± 41.3	186.8 ± 48.9	0.23
TG, mg/dL	147.0 ± 78.5	168.7 ± 100.9	0.03
HDL-C, mg/dL	43.4 ± 11.3	40.4 ± 11.4	0.01
LDL-C, mg/dL	114.1 ± 36.0	108.3 ± 36.1	0.2
AIP	0.49 ± 0.26	0.58 ± 0.27	0.002

Data are expressed as means ± SD. (%). * Including thiazide and indapamide. BMI: body mass index HT: hypertension; DM: Diabetes mellitus; HL: hyperlipidemia; ACE: angiotensin-converting enzyme; ARB = aldosterone receptor antagonist; SGLT2 inhibitor: sodium glucose cotransporter type 2 inhibitor; FPG: fasting plasma glucose; GFR: Glomerular filtration rate TC: total cholesterol; TG: triglyceride level; HDL-C: high-density cholesterol level; LDL-C: low-density cholesterol level; AIP: Atherogenic Index of Plasma. *p* = probability.

**Table 2 diagnostics-13-03249-t002:** Univariable analysis of CAD risk parameters and AIP associated with coronary artery stenosis score (*n* = 190).

Variable	Univariable Analysis		
	OR	(95%CI)	*p*-Value
Age	1.02	(1.0–1.04)	0.04
Gender %	3.20	(1.99–5.16)	<0.0001
BMI kg/cm^2^	1.15	(1.0–1.33)	0.04
HT%	1.69	(1.09–2.6)	0.02
TG mg/dL (1%) *	1.31	(1.02–1.7)	0.03
HDL-C mg/dL	0.98	(0.96–1.0)	0.02
LDL-C mg/dL	0.99	(0.99–1.0)	0.2
AIP	3.74	(1.62–8.64)	0.002

* 1% increase in TG. OR = odds ratio; CI = confidence interval; other abbreviations are the same as those given in Table 1.

**Table 3 diagnostics-13-03249-t003:** Multivariate analysis of AIP for patients with coronary artery stenosis.

Variables	Model 1		Model 2		Model 3	
	OR (95%CI)	*p*-Value	OR (95%CI)	*p*-Value	OR (95%CI)	*p*-Value
Age	1.0 (1.01–1.06)	0.001	1.04 (1.02–1.06)	0.001	1.04 (1.01–1.06)	0.006
Gender %	3.53 (2.1–5.9)	<0.000	3.56 (2.12–5.98)	<0.000	3.6 (2.17–6.18)	<0.000
BMI kg/cm^2^			1.16 (0.99–1.35)	0.06	1.15 (0.99–1.35)	0.06
HT (%)					0.68 (0.42–1.13)	0.1
LDL-C mg/dL			0.99 (0.99–1.0)	0.4	0.99 (0.99–1.0)	0.5
AIP	3.39 (1.41–8.13)	0.003	2.97 (1.2–7.39)	0.02	2.76 (1.1–6.91)	0.03

Model 1: adjusted for age and gender; Model 2: adjusted for age, gender, and BMI; Model 3: adjusted for age, gender, BMI, and HT multicollinearity r > 0.7 (TG-C and HDL-C).

## Data Availability

The data that support the findings of this study are available from the corresponding author, TST, upon reasonable request.

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
