# Peer review of "Atherogenic Index of Plasma Predicts Obstructive Coronary Artery Disease in Patients with Stable Angina Pectoris"

_diagnostics, 2023, doi:10.3390/diagnostics13203249_

Round 1
Reviewer 1 Report
This paper, presented by Kurklu et al., provides a correlation study between AIP and obstructive coronary artery disease in patients with stable angina pectoris. The study confirms the effectiveness of AIP through multifaceted analysis, and provides a full discussion. However, there are a few suggestions and questions:
Abstract
It is recommended to show the basic AIP value of two group;
Mothods
1. Please provide the full name of QCA;
2. The authors said: In order to avoid multicollinearity, parameters with a strong correlation
with AIP (TG, HDL-C r > 0,7) did not enter into the multivariable analysis. Please show the results and put these results as supplemental data.
3. Did the results of TG were fast for 12 hours?
Results
1. The OD in the results section is suggested to be OR;
2. In table 1, please check the LDL-C levels, it showed lower in ≥ 50% Coronary Stenosis Group, but TC levels were higher.
3. Did you correct drug information? How many patients underwent lipid-lowing drugs, like statins? Does statin use could influence the final results?
4. As mentioned above, please show the correlation results.
5. Does LDL-C levels have strong correlation with AIP? If not, please put it into adjust models.
Discussion
1. In the third paragraph of the discussion section, the abbreviation of HDL-C is repeated, it should be Non-HDL-C;
2 .The discussion should be more logical and organized.
Author Response
Response to Reviewer #1
Dear Reviewer
We would like to thank the reviewer for careful reading of our manuscript. We highly appreciate your constructive suggestions and have addressed all the comments. The manuscript has been substantially improved thanks to your careful reading Our response is as follows (the reviewer’s comments are in italics)
This paper, presented by Kurklu et al., provides a correlation study between AIP and obstructive coronary artery disease in patients with stable angina pectoris. The study confirms the effectiveness of AIP through multifaceted analysis, and provides a full discussion. However, there are a few suggestions and questions:
Abstract
Comment 1. It is recommended to show the basic AIP value of two group;
- Response 1. We appreciate the reviewer for pointing this out. We have included the AIP value of the groups in the abstract section please see page 1 (paragraph 1, lines 8-9 )
Mothods
Comment2. Please provide the full name of QCA;
- Response 2. We thank the reviewer. We have added the full name of QCA. Please see page 2 (paragraph 5 line 33 )
Comment 3. The authors said: In order to avoid multicollinearity, parameters with a strong correlation
with AIP (TG, HDL-C r > 0,7) did not enter into the multivariable analysis. Please show the results and put these results as supplemental data.
- Response 3. We appreciate the reviewer’s suggestion. A correlation table was added as an additional data set and showed a strong correlation between AIP, TG, and HDL-C. However, there was no correlation between AIP and LDL-C. Please see page 3 (paragraph 4, line 44) as well as Table S1 page 10 for supplementary data.
Comment 4.. Did the results of TG were fast for 12 hours?
- Response 4. We appreciate the reviewer for pointing this out. All cholesterol results were obtained from fasting blood samples. We have explained this in our manuscript. Please see page3 (paragraph3, lines 23-24 )
Results
Comment 5. The OD in the results section is suggested to be OR;
- Response 5. We apologize for the mistake. We have corrected the OD as OR.
Comment 6. In table 1, please check the LDL-C levels, it showed lower in ≥ 50% Coronary Stenosis Group, but TC levels were higher.
- Response 6. Please accept our apologies for the error. The SD of LDL-C level in the group with ≥50% stenosis has been corrected. Please see Table 1. Despite the fact that LDL-C levels were lower in the ≥ 50% stenosis group, there were no significant differences between the two groups. Furthermore, we expected to observe a lower level of LDL-C in the ≥50% stenosis group. Unfortunately, due to the retrospective nature of the study, we were not able to begin or assess the dosage of statins before the study began.
Comment 7.. Did you correct drug information? How many patients underwent lipid-lowing drugs, like statins? Does statin use could influence the final results?
- Response 7. Thank you very much for pointing this out. It should be noted that only 68 patients were taking statins as the lipid-lowering therapy in the group with <50% stenosis, and 71 patients in the ≥50% stenosis group (Table 1). We were not able to begin lipid-lowering treatment, as we mentioned above. Some of the patients included in our study were already taking statins. In our opinion, many patients should be treated with lipid-lowering medications, but there should be a gap in assessing cardiovascular risk. As well as this, the AIP values can be changed as a result of lipid-lowering treatment and, of course, this can influence the final results. There were, however, no significant differences in statin use or LDL-C levels, which are used as a measure of the effectiveness of a lipid-lowering treatment. According to our hypothesis, AIP may provide additional information regarding non-HDL-C levels, contributing to a better understanding of the dose and titration of lipid-lowering medications. The effectiveness of AIP in the estimation of obstructive coronary artery disease will require prospective studies with a larger number of patients. We have discussed it on page 9 (paragraph 2 lines 27-34 )
- Commnet 8.. As mentioned above, please show the correlation results
- Response 8. We have added as supplementary data Please see Table S1 on page 10
Comment 9. Does LDL-C levels have strong correlation with AIP? If not, please put it into adjust models.
- Thank you for your suggestion. We greatly appreciate it. We have added the LDL-C in the multivariate analysis. Please see pages 5 and 6 Table 3 and page 5 (paragraph1, lines 14-20, and paragraph 2 lines 22-26)
Discussion
Comment 10.. In the third paragraph of the discussion section, the abbreviation of HDL-C is repeated, it should be Non-HDL-C;
- Response 10. Please accept our apologies for the error. It has been corrected
Comment 11. The discussion should be more logical and organized.
- Response 11. Thank you very much for your suggestions; however, due to the number of words required for the special issue we submitted for, we had to provide detailed explanations in the discussion section. To provide a more organized discussion section, we have included and excluded some sentences in some paragraphs. Please see discussion section paragraphs 3-5 and 9.
We truly hope that the revised manuscript is now clearer. We would like to thank the referee again for taking the time to review our manuscript.
Reviewer 2 Report
Dear Authors,
thank you for the opportunity to review the scientific article which is easy to read and perceived very well. My congratulations to you for the wonderful work.
However, if you could show how to find the correlation with other predictors of obstructive CAD besides AIP, the value of your work would increase exponentially.
There are misprints, e.g. Dyslipidaemia (p. 1)
Author Response
Response to Reviewer #2
We would like to thank the reviewer for the careful and thorough reading of this manuscript and for the thoughtful comments and constructive suggestions, which helped to improve the quality of this manuscript. Our response is as follows (the reviewer’s comments are in italics)
Comment1. thank you for the opportunity to review the scientific article which is easy to read and perceived very well. My congratulations to you for the wonderful work.
However, if you could show how to find the correlation with other predictors of obstructive CAD besides AIP, the value of your work would increase exponentially.
Response 1. We greatly appreciate the reviewer's insightful comments. The LDL-C level has been included in both the univariate and multivariate analyses. Furthermore, we have added Table S1 showing the correlation between AIP and LDL-C, TG and HDL-C .
Comment 2.There are misprints, e.g. Dyslipidaemia (p. 1)
Response2. We apologize for the error, it has been corrected
Please accept our sincere thanks for the revised manuscript. Once again, we would like to thank the referees for their time and efforts in reviewing our manuscript. Your assistance in improving the readability of our paper is greatly appreciated.